# Sustainable synthesis of fine chemicals and polymers using industrial chlorine chemistry
Yasuhiro Kohsaka [1,2] ✉, Daisuke Matsuura[3] & Yoshikazu Kimura [3] ✉

To achieve sustainable resource circulation, preparation of reactive species from stable compounds is unavoidable. Chlorine chemistry is an eco-friendly methodology to address this demand. Chlorine is industrially produced from sodium chloride (NaCl), an abundant natural resource in oceans. Chlorine provides various chemical products, including polymers, through chlorination and subsequent conversion reactions. In these reactions, the byproducts are usually hydrogen chloride, which is commercially utilized as hydrochloric acid and is finally neutralized to NaCl after use. Therefore, chlorine chemistry enables fine chemical production from NaCl with almost no wastage. This review provides an overview of the synthesis of fine chemicals and polymers using chlorine chemistry and discusses them from the perspective of sustainability.

Chlorine is a typical element that is abundant in seawater, and organisms on Earth, including humans, use organic chlorine compounds to perform life functions[1]. However, chlorine and organic chlorides are highly toxic, and some have been confirmed to be highly carcinogenic[2]. Given this background, readers may believe that the chlorine industry is the polar opposite of green chemistry and sustainable development. However, the strong toxicity of organic chlorides is inextricably linked to their excellent reactivity of organic chlorides. Organic chlorides are important synthetic intermediates in polymers, pharmaceuticals, and pesticides. In this review, we discuss how the excellent reactivity of organic chlorides contributes to resource circulation and sustainability. For example, improving the reactivity of naturally occurring stable compounds is necessary to prepare fine chemicals from biomass. Similarly, recovery of chemical potential is required for resource circulation from the final products in the chemical industry, such as plastics. In this context, chlorination and functionalization using organic chlorides are effective tools. To consider the potential of chlorine chemistry in the sustainable chemical industry, we first describe the industrial production of chlorine and explain it from the viewpoint of green chemistry.

According to the World Chlorine Council, the production of chlorine in the world was 89 Mt in 2018[3]. The common procedure for chlorine production is electrolysis of a NaCl aqueous solution from sea, which is called the chlor-alkali process. The chlor-alkali process is operated using a mercury cell, diaphragm cell, and membrane cell. Membrane cell electrolysis is currently the most popular process owing to its low toxicity and high energy efficiency. 35% of the

chlorine produced is used in vinyl chloride resins, and 10% is used in bleach, drinking water disinfection, and inorganic chemicals[4]. On the other hand, 25% is used as a raw material for resins, such as propylene oxide, epichlorohydrin, and diisocyanate, and as chlorinated intermediates for pharmaceuticals and agricultural chemicals. In this review, we focus on the production and conversion of organic chlorides into functional chemicals.

Although significant efforts have been made[5,6], chlorine production still consumes significant amounts of electricity. Therefore, chlorine chemistry may not be considered eco-friendly. However, this perception is not always accurate. Japan has the Fuji River, which is a rapid stream that flows around Mt. Fuji to Suruga Bay (Fig. 1). Taking advantage of this topography, six hydroelectric power plants with a total output of 145 MW of Nippon Light Metal were built in the basin, and most of the electricity required for chlorine production is provided by this renewable energy source[7]. Large-scale hydroelectric power generation requires the construction of dams, which may impact the environment and cause water resource problems in river basins. On the other hand, the Fuji River, of which total length is less than 130 km, is a raging river that is listed as one of Japan's three most rapid rivers. Thus, flood control projects have been carried out historically. Currently, the small-scale hydroelectric power plants mentioned above do not simply produce electricity but also contribute to reducing flood damage. Consequently, chlorine production at this plant is a unique business that takes advantage of the characteristics of the local topography, and the issues that are generally discussed on a global scale do not

[1]Research Initiative for Supra-Materials (RISM), Interdisciplinary Cluster for Cutting Edge Research (ICCER), Shinshu University, Nagano, Japan. [2]Faculty of Textile Science and Technology, Shinshu University, Nagano, Japan. [3]Research and Development Department, Iharanikkei Chemical Industry Co. Ltd, Shizuoka, Japan. ✉e-mail: kohsaka@shinshu-u.ac.jp; kimura.yoshikazu@iharanikkei.co.jp

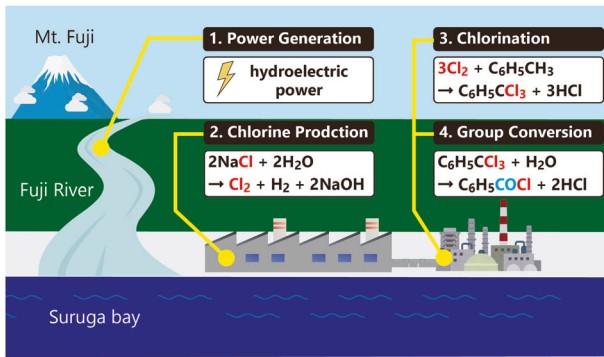

**Fig. 1 | Chlorine chemistry industry along with the Fuji River.** Fine chemicals are produced from chlorine derived via the electrolysis of sodium chloride in Suruga Bay using electricity generated by hydroelectric power from the Fuji River, one of the three fastest rivers in Japan.

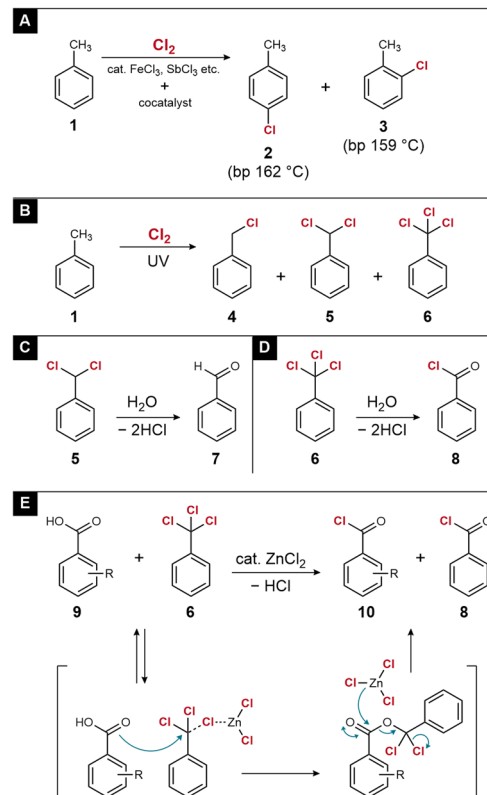

**Fig. 2 | Synthesis of organic chlorides from chlorine. A** Electrophilic aromatic substitution. **B** Photochlorination at the benzyl position. **C** Hydrolysis of benzal chloride to benzaldehyde. **D** Hydrolysis of benzotrichloride to benzoyl chloride. **E** Chlorine–oxygen exchange reaction between carboxylic acid and benzotrichloride.

necessarily apply. The chlorine produced is transported by pipeline to the Iharanikkei Chemical Industry Co., Ltd. plant, where it is converted into various functional chemical products[8]. Furthermore, the by-product, hydrochloric acid, produced in these processes is also used commercially as basic chemical product. Although hydrochloric acid has various demands, including neutralization, cleaning, washing, and acid catalysis in organic reactions, it is ultimately neutralized with a base and processed into sodium or potassium salts. Hence, the chlorine chemistry in this plant guarantees resource circulation. The chemical industry, which takes advantage of these regional characteristics, is important in ensuring sustainability. Therefore, within this complex, chlorine chemistry is an ideal means of adding high value to organic compounds using renewable energy and natural resources, with almost no waste. However, we must refer that surplus hydrogen chloride production has become a problem on a global scale. Thus, chlorine regeneration from hydrogen chloride has attracted attention[9]. In the next section, we introduce a method to produce functional chemical products from chlorine. Sodium hypochlorite (NaClO), another product of the chlorine industry, is a useful reagent for organic synthesis. For more details, please refer to our previous review[10].

## Synthesis of organic chlorides from chlorine
### Chlorination using chlorine

Electrophilic aromatic substitutions[11] and photochlorination[12] are important elementary reactions in the organic chlorine industry. The electrophilic aromatic substitution of toluene (**1**) is not regioselective and affords a mixture of ortho- and para-isomers **2** and **3** (Fig. 2A). As these isomers are useful raw materials for fine chemicals, they must be separated by distillation. Because the difference between the boiling points is only 3 °C at normal pressure, Iharanikkei Chemical Industry purifies these isomers using a distillation column with more than 300 theoretical plates. In contrast, the photochlorination is regioselective. Because the reaction proceeds via a radical mechanism, it occurs regioselectively at certain positions to afford stable radicals. For example, the photochlorination of toluene occurs at the benzyl position (Fig. 2B). However, photochlorination is a chain reaction in which a mixture of mono-, di-, and tri-substituted toluenes **4**, **5**, and **6** is generated. These compounds are separated using a distillation column with more than 30–40 theoretical plates.

### Di- and tri-chlorotoluene derivatives

The products of toluene photochlorination can be converted to other functional chemicals via further modification. **4** is majorly used as a benzylation reagent in organic chemical industry. The hydrolysis of α,α-dichlorotoluene (**5**), known as benzal chloride, affords benzaldehyde **7**[13]. For example, a patent of performing hydrolysis at 70–75 °C in the presence of a zinc hydroxide catalyst was reported[14]. This is a

typical industrial process for the preparation of aromatic aldehydes (Fig. 2C). Similarly, the hydrolysis of α,α,α-trichlorotoluene (**6**), also known as benzotrichloride, results in benzoyl chloride (**8**) (Fig. 2D). This reaction can proceed simply by adding water and heating; however, the hydrolysis of **8** also proceeds to afford benzoic acid if excess water is added[15]. Therefore, water must be added gradually to suppress excessive reactions. Industrially, a catalyst should be added to increase the reaction rate, and catalyst residues should also be considered. $ZrO_2 \cdot nH_2O$[16], $FeCl_3$[17], and $ZnCl_2$[18] have been reported as suitable catalysts. Moreover, the chlorine–oxygen exchange reaction between **6** and aromatic carboxylic acid **9** affords two aromatic acyl chlorides, **10** and **8** (Fig. 2E). In contrast to the hydrolysis of **8**, which may afford benzoic acid via an excessive reaction, the chlorine-oxygen exchange does not accompany such side reactions. Therefore, it has been widely studied industrially, and many related patents using $ZnCl_2$[19] and $FeCl_3$[20] catalysts have been filed.

The combination of the elementary reactions described in Fig. 2 enables the synthesis of various organic chlorides with higher values. For example, toluene is converted to *p*-chlorobenzaldehyde (**12**), an intermediate of pharmaceuticals and agricultural chemicals, via an electrophilic aromatic substitution reaction, followed by photochlorination and hydrolysis (Fig. 3A)[21]. Terephthaloyl chloride (**16**), a monomer of polyesters and polyamides, is prepared from *p*-xylene via photochlorination and a chlorine–oxygen exchange reaction (Fig. 3B)[22]. This protocol is also effective for the synthesis of 1,4-cyclohexane dicarbonyl chloride (**18**), a promising monomer for bio-based polyesters (Fig. 3C)[23]. Isomerization from the cis isomer to the trans isomer occurs in the chlorine–oxygen exchange reaction between **17** and **6**. Nevertheless, because the melting points of **18 T** and **18 C** are significantly different, separation by filtration resulted in high purity of each isomer[24].

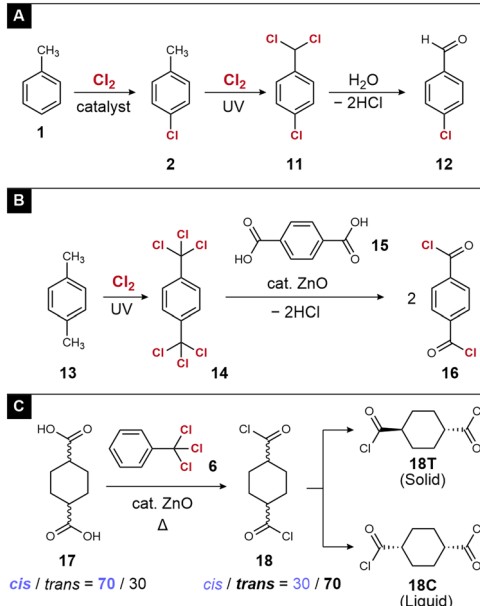

**Fig. 3 | Selected examples of fine chemical synthesis from chlorine. A** Synthesis of 4-chlorobenzaldehyde from toluene. **B** Synthesis of terephthaloyl chloride from *p*-xylene. **C** Synthesis and separation of 1,4-cyclohexane dicarbonyl chloride.

## Phthaloyl chloride

As described later in this section, phthaloyl chloride (**24**) is a convenient reagent for chlorination and oxidation. Because **16** was prepared from **13** (Fig. 3B), a similar procedure seems effective for the preparation of **24**. However, the photochlorination of *o*-xylene (**19**) does not afford the hexa-substituted product because the steric hindrance of penta-substituted product **20** prevents further photochlorination (Fig. 4A)[25]. Therefore, a multistep reaction is required to obtain **24**[26]. The hydrolysis of **20** resulted in aldehyde **21**, which is in a tautomerization with lactone **22**. Photochlorination of **22** affords **23**, which is a tautomer of **24**. This multistep synthesis is inefficient, and other routes to **24** are desirable.

Treatment of phthalic anhydride (**25**) with common chlorination reagents, such as PCl₅ and SOCl₂, yielded **24**, although unreacted **25** remained in the reaction system (Fig. 4B)[27,28]. Unfortunately, the boiling point of **24** (276 °C) is close to that of **25** (284 °C), and purification by distillation is not practical. Thus, quantitative chlorination was necessary to obtain **24** with high purity. Some patents claim that phosgene is effective in addressing this demand, although its use should be avoided in industrial applications owing to its toxicity[27,29,30]. Kyrides reported the chlorination of **25** using benzotrichloride **6** in the presence of ZnCl₂ (Fig. 4C)[31]. Technical issues include the requirement of a large amount of catalyst (10 mol%) for the reaction at 120 °C and inadequate purity (< 95%) due to the remaining unreacted **25**. We discovered that ZrCl₄ was a more active catalyst than ZnCl₂[32]. A small amount (0.1 mol%) of ZrCl₄ was sufficient even at 160 °C,

**Fig. 4 | Synthesis of phthaloyl chloride (24) and its application to chlorination and formylation reactions. A–C** Synthesis. **D–G** Functionalization using **24** and its derivatives.

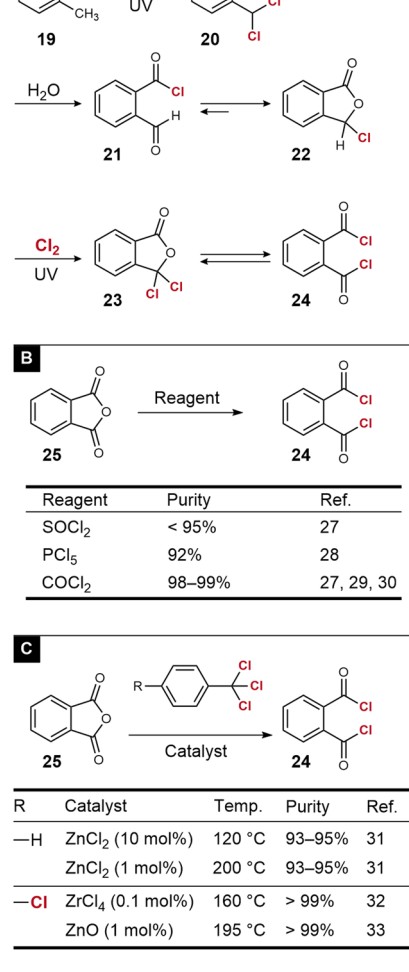

| Reagent | Purity | Ref. |
|---|---|---|
| SOCl₂ | < 95% | 27 |
| PCl₅ | 92% | 28 |
| COCl₂ | 98–99% | 27, 29, 30 |

| R | Catalyst | Temp. | Purity | Ref. |
|---|---|---|---|---|
| —H | ZnCl₂ (10 mol%) | 120 °C | 93–95% | 31 |
|  | ZnCl₂ (1 mol%) | 200 °C | 93–95% | 31 |
| —Cl | ZrCl₄ (0.1 mol%) | 160 °C | > 99% | 32 |
|  | ZnO (1 mol%) | 195 °C | > 99% | 33 |

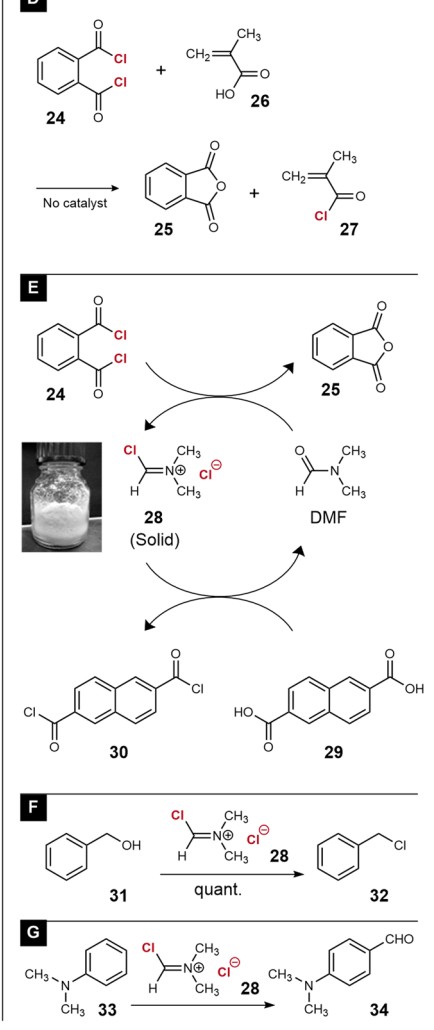

resulting in **24** with very high purity (>99%). ZnO was also found effective, although the activity was lower than ZrCl₄[33].

**24** functions an excellent chlorination reagent for carboxylic acids (Fig. 4D)[34,35]. Chlorination does not require catalysts. In fact, the treatment of methacrylic acid (**26**) with **24** resulted in **27** with a high purity (> 99%)[36]. Such high-purity acyl chlorides are useful for precise polymerization chemistry[37]. **24** functions as a precursor of the Vilsmeier-Haack (V-H) reagent (Fig. 4E). For example, the treatment of **24** with *N,N*-dimethylformamide (DMF) in a low-polarity solvent such as toluene, *o*-chlorotoluene, or 1,4-dioxane at ambient temperature resulted in the precipitation of the V-H reagent **28**[38]. Filtration of the reaction mixture afforded **28** as a colorless solid, and the reaction with carboxylic acid afforded the corresponding acyl chloride. For example, the reaction of **28** and **29** in 1,4-dioxane at 70 °C afforded **30** in a high yield (86%). Note that **30** was insoluble in 1,4-dioxane and DMF at 25 °C, and colorless precipitation was observed after cooling the reaction mixture. Thus, **30** was obtained by filtration. **28** was also effective for the chlorination of primary and secondary alcohols[38] (Fig. 4F) and formylation of aromatic rings[38,39] (Fig. 4G).

## Sustainable polymers using monomers derived via chlorine chemistry

As mentioned above, chlorine chemistry provides aromatic aldehydes and acyl chlorides with almost no waste. Historically, dialdehydes have been applied to polyaddition via imine formation[40,41] and the Morita-Baylis-Hillman reaction[42,43], whereas divalent acyl chlorides are common monomers of polyesters and nylons. On the other hand, monomers derived via chlorine chemistry, including dialdehydes and divalent acyl chlorides, have gained researchers' attention from a viewpoint of sustainability. This section introduces the new trends in the field of polymer chemistry.

### Bio-based divalent acyl chlorides

**2,5-furandicarboxy chloride.** The production of plastics has been supported by the petrochemical industry, although a shift to bio-based materials is desirable. 2,5-Furandicarboxylic acid, derived from glucose or fructose[44], is one of the top value-added chemicals in biomass, as defined by the US Department of Energy[45]. Therefore, the corresponding acyl chloride **35** is important as a monomer with higher electrophilicity.

Wu and coworkers prepared a bio-based polyester **38** via acyclic diene metathesis (ADMET) polymerization of diene **37**, derived from **35**, followed by hydrogeneration (Fig. 5A)[46]. Notably, the physical properties of **38** were similar to those of polyethylene. The ring-opening polymerization of cyclic oligoester **40** prepared from **35** and diol **39** has also been reported (Fig. 5B)[47]. As the direct polycondensation of divalent carboxylic acids and diols does not afford a high-molar-mass polymer[48], such a two-step process is often preferred. Recently, Arai et al. reported the synthesis of polyamide **43** from **35** and bifuran diamine **42**[49]. This is a fully furan-based polymer with a high glass transition temperature ($T_g$) of 193 °C (Fig. 5C). Kanetaka et al. prepared poly(ether ketone) **47** from **35** (Fig. 5D)[50]. **47** exhibited a 5% weight loss temperature ($T_{d5}$), melting point ($T_m$), and $T_g$ of 414, 337, and 143 °C, respectively, suggesting its performance as an engineering plastic, although the content of bio-based moieties was not high.

**Fig. 5 | Bio-based divalent acyl chlorides and their polymerization. A–E** Polymer synthesis using furan-2,5-carbonyl dichloride (**35**). **F** Synthesis of itaconyl chloride (**58**).

Epoxy resins account for the majority in the thermosetting resin market and are used in a variety of applications, including coatings and fiber-reinforced plastics, because of their thermal stability, mechanical strength, and chemical resistance. It is desirable to replace monomers with biomass-derived products without sacrificing their excellent properties. Miao et al. reported a new epoxy resin prepared from **35**, eugenol, and methyl hexahydrophthalic anhydride[51]. Although almost all moieties of this epoxy resin were composed of bio-based moiety, it exhibited excellent mechanical performance and heat resistance comparable to a common petroleum-derived epoxy resin. Recently, Wu et al. reported a bio-based and chemically recyclable epoxy resin (Fig. 5E)[52]. Diepoxide **48**, prepared from **35**, and bio-based diamine **49** were reacted to form a network polymer **50**. Epoxy resin **50** exhibited a $T_g$ of 170 °C, a storage modulus ($E'$) of 1.2 GPa, and chemical resistance alcohols except methanol. Remarkably, the alcoholysis of **50** using methanol at 70 °C resulted in the recovery of diester **51** and alcoholic residues, which could be converted to **49** and glycidol, the raw materials of **50**. Therefore, this resin was evaluated to be chemically recyclable.

**Itaconyl chloride**. Itaconic acid[53] (**56**) is another dicarboxylic acid listed in the top value-added chemicals in biomass by the US Department of Energy[45]. Its derivatives have also been recognized as bio-based acryl monomers for radical polymerization[54–56]. However, the corresponding acyl chloride (**58**) was difficult to synthesize using common chlorination reagents such as $SOCl_2$ and $PCl_5$ in high yield and purity, because the chlorination of **56** afforded a stable intermediate, itaconyl anhydride (**57**)[57]. Because polycondensation is sensitive to monomer purity,

a synthetic procedure to prepare **58** with a high purity (>99%) is desirable.

Dichloromethyl alkyl ethers function as strong chlorination reagents for carboxyl acids and esters in the presence of Lewis acid catalysts, such as $ZnCl_2$. Because dichloromethyl alkyl ethers with short (C1–C3) alkyl groups have low boiling points, butyl ether **55** was prepared via the reaction of butyl formate and oxalyl chloride (Fig. 5F)[58,59]. The DMF catalyst was not effective, probably because of the poor solubility of the corresponding intermediate, V-H reagent **28**. In contrast, N-methyl-N-phenyl formamide (**54**) was effective in producing **55**. The treatment of **56** or **57** with **55** in the presence of $ZnCl_2$ or ZnO resulted in a high purity (>99%) and quantitative yield of **58**[60]. In the polycondensation of **56** and its esters with diols, the thermal polymerization of acryl moieties via a radical mechanism must be avoided to obtain linear polymers. Thus, the high reactivity of **58** was effective for achieving polycondensation at lower temperatures.

**Chemically recyclable monomers derived via chlorine chemistry**
**Phthalaldehydes**. As mentioned at the beginning of this section, polyaddition of dialdehydes and diamines are reversible reaction, and the resulting polyimines, also known as "dynamers," have a potential to chemical recycling[40]. Recently, Xiao et al. reported polycondensation using the Hooz reaction, a three-component condensation of divalent diazocarbonyl compounds **59**, dialdehydes **60**, and trialkylborons **61** (Fig. 6A)[61]. Although the main chain was composed of a carbon skeleton, the retro-aldol reaction using potassium hydroxide resulted in a recovery of **60**. Thus, this polymer was partially recyclable.

**Fig. 6 | Chemically recyclable monomers prepared via chlorine chemistry.**
**A** Polymerization via Hooz reaction using dialdehyde (**60**). **B** Cationic polymerization of phthalaldehyde (**63a**) and its derivatives. **C** Polymerization and multi-step chemical recycling of cyclic ketene acetal ester **66**. **D** Polymerization and multi-step chemical recycling of cyclic vinyl ester **76**.

Phthalaldehyde (**63b**) affords polyacetals via cationic polymerization, although the ceiling temperature ($T_c$) is low ($-36\,°C$)[62]. Therefore, the activation of the chain end immediately causes depolymerization at ambient temperature, and this feature is applied to chemically amplified photoresists[63] and self-immolative polymers[64], that is, polymers that undergo depolymerization triggered by the removal of specific groups usually located in chain ends or pendant groups[65,66]. In contrast, the cyclic polymer **64** was relatively stable because of the absence of chain ends (Fig. 6B). Lutz et al. investigated the effects of the aromatic substituents of **63** on their $T_c$[67]. Notably, the degradation temperature ($T_d$) exhibited a linear relationship with $T_c$, and the monomer with a perfluoro aromatic ring (**63e**) exhibited thermal stability up to ca. 200 °C. Therefore, stability and recyclability can be tuned by modifying the aromatic substituents.

**Cyclic viny esters**. Cyclic vinyl esters are another category of monomers derived by chlorine chemistry. For example, cyclic ketene acetal ester **66** was obtained via the chlorination of acetylsalicylic acid (**72**) by V-H reagent **28** followed by intramolecular esterification (Fig. 6C)[68]. Kazama et al. reported radical polymerization of **66** using 2,2′-azobis(isobutyronitrile) (AIBN)[69,70]. Notably, hydrolysis of the resulting polymer **67** in dimethyl sulfoxide (DMSO) using HCl aq or NaOH aq afforded acetic acid (**68**) and salicylic acid (**69**). The reaction mechanism was explained by the hydrolysis of each component consisting of **67**, that is, esters, acetals, and 1,3-dicarbonyl skeletons. Because **68** and **69** are the raw materials of **72**, polymers **67** and **71** are principally chemically recyclable. In practice, the reuse of **68** is not realistically effective; it is not worth reusing **68** because of the hassle of isolation and purification. The true value of these results is that **69** was used as a platform to construct a polymerization-active vinylidene group using acetic acid. **69** was recovered by extraction of hydrolysis products of **67**. Thus, the resynthesis of **72** was possible using the recovered **69** and the renewable reagent, acetic anhydride. In this meaning, **67** could be evaluated as a chemically recyclable vinyl polymer. Recently, Goto et al. reported similar cyclic ketene acetal esters as recyclable vinyl monomers[71]. Kazama et al. also reported that the cationic polymerization of **66** using BF$_3$·OEt$_2$ as an initiator also afforded **67**[70]. In contrast, the use of a binary initiator of **70** and EtAlCl$_2$ below 0 °C resulted in poly(ester ketone) **71** by ring-opening polymerization (ROP) via cation isomerization. **71** was also degraded by NaOH aq to yield **68** and **69** via the hydrolysis of esters and 1,3-dicarbonyl skeleton, respectively. Therefore, **66** is a recyclable two-role monomer for vinyl polymerization and ROP.

Cyclic vinyl ester **76**, which was prepared from **25** and malonic acid (**73**), is another recyclable vinyl monomer (Fig. 6D)[72]. Herein, **25** and **53** are products of chlorine chemistry. Radical polymerization of **76** resulted in the corresponding vinyl polymer **77**[73,74]. The hydrolysis of **77** causes the ring-opening of lactone moieties to enhance the steric repulsion of pendant groups, leading to chain scission and depolymerization to afford monomer precursors **74** and **75** as a mixture of tautomers[72].

## Covalent adaptable networks

Molecular strategies that utilize reversible covalent bond formation are known as dynamic covalent chemistry (DCC)[75]. DCC is attractive for achieving reversible polymer synthesis/degradation[76]; the dynamers[40] and polyacetals described in 3.2.1 are typical examples. Similarly, DCC has been applied to the reversible crosslinking of polymers[77]. Historically, the aim has been to completely decrosslink network polymers and recycle recovered linear polymers. In recent years, attention has shifted from decrosslinking to the reconstruction of crosslinked structures. If the exchange of the crosslinked points is sufficiently fast, the reformation of the crosslinked structures follows deformation. Thus, the crosslinked resin becomes processable, even though the crosslinked structure is not removed. Such resins are known as covalent adaptable networks (CANs)[78,79]. CANs, whose crosslinked structure is reformed via bond exchange by an addition–elimination mechanism (associative bond exchange), are specifically defined as vitrimers because of their unique rheological properties[80]. In addition to processability, CANs are chemically recyclable via decrosslinking. Therefore, CANs are expected to contribute to sustainability as an alternative to conventional thermosetting resins.

Condensation between amines and aldehydes, that is, imine formation, is a reversible reaction. Importantly, the imide exchange reaction, which is typically promoted by heat and polar solvents, proceeds even in the absence of catalysts. Therefore, imine bonds are convenient for application in CANs. Terephthalaldehyde (**60a**), a product of α,α,α′,α′-tetrachloroxylene hydrolysis, is often used for imine formation in CANs. For example, Ling et al. prepared CANs (vitrimers) by crosslinking poly(amide imine) **80** with amine ends using triester **81** (Fig. 7A)[81]. The resulting network polymer **82** exhibited stress relaxation and processability, suggesting the reconstruction of the network structure by an imine exchange reaction. The tensile strength was up to 47 MPa, and the creep resistance was improved compared with that of typical imine-based CANs. These properties were attributed to the hydrogen bonds formed in the amide groups.

Recently, Deng et al. reported CANs prepared from acylhydrazine **83** and di- or tri-aldehydes **60a** and **84** (Fig. 7B)[82]. In this study, the 1,2-dithiolane moieties of **83** were incorporated into ROP to construct the backbone. Resin **85** exhibits both processability and mechanical toughness owing to the imine exchange reaction and hydrogen bonds. In addition, depolymerization of the backbone polymer occurred in dimethyl sulfoxide (DMSO). Additional experiments suggested that depolymerization was promoted in polar solvents, such as DMSO and DMF, and the free acylhydrazine groups functioned as initiators of depolymerization. Therefore, the combination of DMSO for depolymerization and hydrazine for decrosslinking of imine bonds afforded **83**, indicating partial chemical recycling.

Aromatic aldehydes are good reactants for Knoevenagel condensation. Thus, Wang et al. used aromatic aldehydes as good reactants for the Knoevenagel condensation of **60a** and triarm α-cyanoacetate **86**[83]. Model reactions using small molecules suggested bond exchange between the Knoevenagel adducts. Network polymer **87** exhibited malleability and reprocessability, suggesting its performance as a CANs. Typically, CANs have poorer mechanical properties, particularly creep resistance, than conventional thermosetting resins. To improve them, hydrogen bonds were incorporated in the previously described examples **82** and **85**. In contrast, **87** exhibited excellent mechanical properties despite the absence of hydrogen bonds. The Young's modulus and tensile strength were 3.8 GPa and 102 MPa, respectively, while the coefficient of thermal expansion from 30 to 80 °C was 82 ppm K$^{-1}$. These excellent properties were attributed to the long conjugated system of Knoevenagel adducts. Furthermore, the treatment of **87** with sodium hydroxide resulted in the recovery of **60a**.

## Outlook

This review provides an overview of the production of chlorine and organic chlorides and their applications in sustainable polymers. Since chlorine gas is rarely used in a laboratory, researchers may have an impression of its toxicity and corrosivity to chlorine and organic chlorides. We hope to improve the evaluation of chlorine chemistry through this review. The benefits of chlorine chemistry can be summarized as follows:

1. Chlorine is an abundant naturally derived resource. Chlorine is an abundant, naturally derived resource that can be obtained through the electrolysis of seawater. The power consumption problem can be almost completely solved by hydroelectric power generation using rapids. A chemical industry that takes advantage of regional characteristics is important to promote SDGs.

**Fig. 7 | Terephthalaldehyde (60a) affords covalent adaptable networks (CANs). A** CANs utilizing imine exchange. **B** CANs utilizing hydrazine imine exchange and disulfide exchange. **C** CANs utilizing Knoevenagel adduct exchange.

2. A clean chemical industry with almost no waste is produced. Aromatic aldehydes and acyl chlorides are produced from toluene derivatives, chlorine, water, and ultraviolet (UV) light. Acyl chloride can be converted into other acyl chlorides via oxygen-chlorine exchange reactions. The hydrochloric acid produced as a byproduct also has commercial value.

3. Chlorine chemistry enables the use of bio-based compounds and resource circulation. Acyl chloride, aldehyde, hydrochloric acid, and sodium hydroxide, produced by chlorine chemistry, are highly reactive reagents. These reagents are effective for chemical modification and activation of stable and unreactive biomass-derived compounds. In addition, resource recycling through synthesis/degradation using these reagents has been studied in cutting-edge polymer chemistry.

We believe that chlorine chemistry will lead to green chemistry and the development of sustainable polymer materials. However, acyl chlorides are easily hydrolyzed to generate hydrogen chloride during transportation and storage, which poses the problem of corroding metal products. To address this issue, we examined acyl-1,2,4-triazole **89**, derived from the corresponding acyl chloride **88** and 1,2,4-triazole (Fig. 8)[84]. In the presence of 4-(*N,N*-dimethylamino)pyridine (DMAP), **89** converted alcohols to esters at a rate comparable to that of **88**. The byproduct was 1,2,4-triazole, which is water-soluble and has low toxicity. Bulk polycondensation using divalent acyl-1,2,4-triazole **91** and diol **92** at 80 °C afforded polyester **93**[85,86].

Sustainable use of organic chlorides requires safe and convenient alternatives that can be easily derived from organic chlorides. Furthermore, the huge amount of electricity consumed during chlorine production is a significant issue, even though it is managed by hydroelectric power. Improvements in chlorine production methods remain an important research topic. We look forward to the day when chlorine chemistry can overcome these issues and truly contribute to the SDGs.

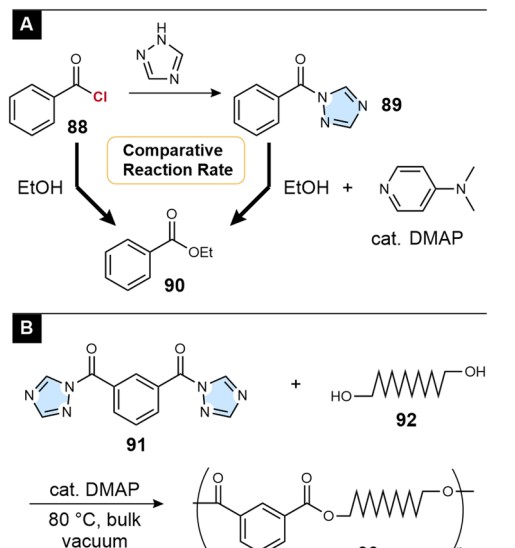

**Fig. 8 | Acyl 1,2,4-triazoles as an alternative of acyl chloride. A** Esterification using benzoyl chloride (**88**) and benzoyl -1,2,4-triazole (**89**). **B** Polycondensation using divalent acyl-1,2,4-triazole **91**.

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

## Acknowledgements

The authors thank to Ms. Shiori Sato to provide information on a chlorine market.

## Author contributions

Y.Ko. defined the scope of the review, wrote sections, prepared figures, edited, and proofread. D.M. provided information on a chlorine market. Y.Ki. provided information on chlorine chemistry and industry, and contributed proofreading.

## Competing interests

Y.Ki. and D.M. report a relationship with Iharanikkei Chemical Industry Co., Ltd. that includes: employment. Y.Ko. declare that they have no known competing financial interests or personal relationships that could have appeared to influence the work reported in this paper.
