## [Peer Review file · Communications Chemistry]

Sustainable Synthesis of Fine Chemicals and Polymers Using Industrial Chlorine Chemistry

Corresponding Author: Professor Yasuhiro Kohsaka

Version 0:

Reviewer comments:

Reviewer #1

(Remarks to the Author)

The manuscript is relevant, and contributes with the field of sustainable chemistry and circularity. My only recommendation is to include a paragraph in the introduction (1. Chlorine: Production and Market) on what has already been published globally on the greener and more sustainable production and sustainable use of chlorinated compounds, in addition to refs 1-4. After the introduction of this paragraph, the manuscript can be considered for publication.

Reviewer #2

(Remarks to the Author)

The authors present a review that shows a route from sodium chloride to interesting materials, mainly through the use of acyl chlorides, which in turn are obtained by photochlorination at the benzyl position. The title "A Gift from Ocean: Chlorine Chemistry for the Production of Fine Chemicals and their Application to Sustainable Polymer Materials" does not reflect the content well. I would have expected to learn more about chlorination, whereas the paper mainly deals with photochlorination for the production of benzotrichloride and its derivatives and their use as reagents for the synthesis of acyl chlorides. The title should be changed to reflect the importance of acyl chlorides in the article. Otherwise, the article does a good job of presenting the journey of a chlorine atom from its original source as a "salt" to the production of new materials. However, some information on the recycling of waste HCl would also be informative and would fit well with the aim of the article. This is all the more true when HCl is reused in these processes. The examples of the use of acyl chlorides are well presented, including in the wide range of applications, from the classic phthaloyl chlorides to the bio-based furandicarboxyl chlorides and more complex polymers. I would recommend that the paper is accepted for publication after authors addresses the comments.

Some further remarks:

- Preface, I21: "the byproducts are usually chloride salts such as NaCl." In reality, the by-product is usually HCl, since chlorine usually replaces a hydrogen atom.
- P1, I38: Although hydropower plants are considered a renewable energy source, they have a very high environmental impact. It would be better to say that the industry mitigates its high energy consumption in chlorine production by using energy from renewable sources.
- Figure 2: Nothing is said about the separation of products of the photochlorination of toluene. What about the use of benzyl chloride?
- HCl is a by-product in these reactions. Is HCl reused? It would be instructive to read about this.
- Figure 3C (and related reactions): Benzotrifluoride is used to make acyl chlorides like 18. Nothing is said about the by-product that is formed from the benzotrifluoride. What is it and how is it treated (separation, reuse, regeneration)?
- P3, I113: "A trace amount (0.1 mol%)...". I doubt that colleagues in analytical chemistry would define 0.1 mol% as a trace amount. It should be replaced by "i.e. small".
- P4, I134: The name should be furan-2,5-carbonyl dichloride.
- P4, I165: 56 should be deleted.

Version 1:

Reviewer comments:

Reviewer #1

(Remarks to the Author)

The authors answered all the reviewers' questions adequately and made significant changes to the manuscript. I recommend its publication.

Reviewer #2

(Remarks to the Author)

The authors have modified the manuscript according to the reviewers' comments. I recommend the article for publication.

Response to Reviewers

Reviewer-1

The manuscript is relevant, and contributes with the field of sustainable chemistry and circularity.

Response: Thank you very much for your reviewing and high evaluation of our manuscript.

My only recommendation is to include a paragraph in the introduction (1. Chlorine: Production and Market) on what has already been published globally on the greener and more sustainable production and sustainable use of chlorinated compounds, in addition to refs 1-4. After the introduction of this paragraph, the manuscript can be considered for publication.

Response: Thank you for your pointing out. Since the 'Preface' has been substituted 'Abstract' according to the direction of Authors' Checklist, the first section become more abrupt. This makes your suggestion more important. Unfortunately, chlorine chemistry and industry have scarcely described from a viewpoint of green chemistry and sustainability.

The majority of papers raise concerns about the toxicity and environmental pollution of organic chlorides. Several papers have been published that refute this assertion by discussing the role of organic chlorides in nature. However, no proposal has been made to use chlorine chemistry as a means to achieve green chemistry or sustainability. This is why this review is groundbreaking. We have added a paragraph at the beginning to explain such background and the purpose of this review, which I believe makes the introduction easier to read.

In addition to your comment, we have changed the citation of ref. 17-19 for more suitable paper and patents.

Reviewer-2

The authors present a review that shows a route from sodium chloride to interesting materials, mainly through the use of acyl chlorides, which in turn are obtained by photochlorination at the benzyl position. The title "A Gift from Ocean: Chlorine Chemistry for the Production of Fine Chemicals and their Application to Sustainable Polymer Materials" does not reflect the content well. I would have expected to learn more about chlorination, whereas the paper mainly deals with photochlorination for the production of benzotrichloride and its derivatives and their use as reagents for the synthesis of acyl chlorides. The title should be changed to reflect the importance of acyl chlorides in the article.

Response: Thank you for your pointing out. Because the chlorination procedure varies depending on the substances in the synthesis of acyl chlorides, the volume on acyl chloride synthesis has become larger. Therefore, the synthesis of aromatic aldehydes from benzal dichloride, which is another main topic of this Review but can be achieved simple hydrolysis, become faded. Thus, we consider that the spotlight must be on both acyl chlorides and aldehydes. In addition, the initial title of 'chlorine chemistry' may mislead the readers' as if this Review focused on chlorine production or inorganic chlorides. For these reasons, we have corrected the title as follows:

A Gift from Ocean: **Organic** Chlorine Chemistry for the **Industrial** Production of **Aldehydes and Acyl chlorides** and their Application to Sustainable Polymer Materials

Otherwise, the article does a good job of presenting the journey of a chlorine atom from its original source

as a “salt” to the production of new materials.

Response: Thank you very much for your high evaluation of our manuscript.

However, some information on the recycling of waste HCl would also be informative and would fit well with the aim of the article. This is all the more true when HCl is reused in these processes.

Response: At the chlorine chemistry plant along with the Fuji River introduced in this review, all of the hydrogen chloride by-product is commercialized as hydrochloric acid. Although the sold hydrochloric acid is used for various purposes, such as neutralization, cleaning, washing, and acid catalysis, it is ultimately neutralized with a base and processed into sodium or potassium salts. Hence, this commercial model guarantees the resource circulation. However, it is also fact that the surplus hydrogen chloride production has become a problem on a global scale. Thus, the regeneration of chlorine from hydrogen chloride has gained attention. To explain this background, the following sentences and a reference have been added to the third paragraph of the introduction in the revised manuscript.

Furthermore, the by-product, hydrochloric acid, produced in these processes are also used commercially as basic chemical products. Although hydrochloric acid has various demands, including neutralization, cleaning, washing, and acid catalysis in organic reactions, it is ultimately neutralized with a base and processed into sodium or potassium salts. Hence, the chlorine chemistry in this plant guarantees resource circulation. The chemical industry, which takes advantage of these regional characteristics, is important in ensuring sustainability. Therefore, within this complex, chlorine chemistry is an ideal means of adding high value to organic compounds using renewable energy and natural resources, with almost no waste. However, we must refer that surplus hydrogen chloride production becomes a problem on a global scale. Thus, chlorine regeneration from hydrogen chloride has attracted attention.⁹

- Preface, I21: “the byproducts are usually chloride salts such as NaCl.” In reality, the by-product is usually HCl, since chlorine usually replaces a hydrogen atom.

Response: Thank you for your pointing out. The sentence has been revised as follows:

In these reactions, the byproducts are usually hydrochloride, which is commercially utilized as hydrochloric acid and is finally neutralized to NaCl after use.

- P1, I38: Although hydropower plants are considered a renewable energy source, they have a very high environmental impact. It would be better to say that the industry mitigates its high energy consumption in chlorine production by using energy from renewable sources.

Response: We agree that chlorine production requires a huge amount of electricity and that this needs to be improved. The original manuscript also mentions this issue at the beginning of this paragraph. However, please note that the latter part of the paragraph does not refer to chlorine production in general, but focuses on that in the Fuji River basin. In fact, almost all of the electricity consumed at this plant is generated by hydroelectric power. However, as you pointed out, it is fact that large-scale hydroelectric power generation involving the construction of dams causes problems with high environmental impact. We must explain that the hydroelectric power generation on the Fuji River is different from such large-scale hydroelectric power generation. Since the Fuji River is originally a short and violent river, measures to reduce flooding by installing small dams have been developed historically. In modern times, the small dams are also applied to small-scale hydroelectric power generation for chlorine production. Similar models are being implemented in other parts of Japan (<https://www.asahi-kasei.co.jp/bemliese/en/what-bemliese/sustainable.html>). We believe that

chemical industries that take advantage of the characteristics of these regions will be a breakthrough in solving global issues. In the revised version, we have revised the article to explain in detail these unique Japanese circumstances while referring to the environmental issues related to hydroelectric power generation, as follows:

Large-scale hydroelectric power generation requires the construction of dams, which may impact the environment and cause water resource problems in river basins. On the other hand, the Fuji River, of which total length is less than 130 km, is a raging river that is listed as one of Japan's three most rapid rivers. Thus, flood control projects have been carried out historically. Currently, the small-scale hydroelectric power plants mentioned above do not simply produce electricity but also contribute to reducing flood damage. Consequently, chlorine production at this plant is a unique business that takes advantage of the characteristics of the local topography, and the issues that are generally discussed on a global scale do not necessarily apply.

- Figure 2: Nothing is said about the separation of products of the photochlorination of toluene. What about the use of benzyl chloride?

Response: Thank you for your pointing out. The following sentence has been added at the last of 2.1:

These compounds are separated using a distillation column with more than 30–40 theoretical plates.

Furthermore, the following sentence has been added to 2.2.

4 is majorly used as a benzylation reagent in organic chemical industry.

- HCl is a by-product in these reactions. Is HCl reused? It would be instructive to read about this.

Response: Thank you for your suggestion. As answered above, we have described the use of HCl in the Introduction. Because of the limitation of length, we do not refrain this in 2.1 and 2.2.

Furthermore, the by-product, hydrochloric acid, produced in these processes are also used commercially as basic chemical products. Although hydrochloric acid has various demands, including neutralization, cleaning, washing, and acid catalysis in organic reactions, it is ultimately neutralized with a base and processed into sodium or potassium salts. Hence, the chlorine chemistry in this plant guarantees resource circulation. The chemical industry, which takes advantage of these regional characteristics, is important in ensuring sustainability. Therefore, within this complex, chlorine chemistry is an ideal means of adding high value to organic compounds using renewable energy and natural resources, with almost no waste. However, we must refer that surplus hydrogen chloride production becomes a problem on a global scale. Thus, chlorine regeneration from hydrogen chloride has attracted attention.⁹

- Figure 3C (and related reactions): Benzotrifluoride is used to make acyl chlorides like 18. Nothing is said about the by-product that is formed from the benzotrifluoride. What is it and how is it treated (separation, reuse, regeneration)?

Response: Since we never describe the chemistry of benzotrifluoride, we answer as your comment is on benzotrichloride. The original reaction has been explained in the previous paragraph using Figure 2 (for example, the preparation of 18 is based on Figure 2E). In Figure 2, we have described the by-product of HCl. We have referred the use of HCl in the Introduction.

- P3, I113: "A trace amount (0.1 mol%)...". I doubt that colleagues in analytical chemistry would define 0.1 mol% as a trace amount. It should be replaced by "i.e. small".

Response: Thank you very much for your suggestion. We have revised it according to your suggestion.

- P4, I134: The name should be furan-2,5-carbonyl dichloride.

Response: Thank you very much for your suggestion. We have revised it according to your suggestion.

- P4, I165: 56 should be deleted.

Response: Thank you very much for your suggestion. We have revised it according to your suggestion

In addition to your comment, we have changed the citation of ref. 17-19 for more suitable paper and patents.